# Ginkgolide B Regulates CDDP Chemoresistance in Oral Cancer via the Platelet-Activating Factor Receptor Pathway

**DOI:** 10.3390/cancers13246299

**Published:** 2021-12-15

**Authors:** Kohei Kawasaki, Atsushi Kasamatsu, Toshiaki Ando, Tomoaki Saito, Takafumi Nobuchi, Ryunosuke Nozaki, Manabu Iyoda, Katsuhiro Uzawa

**Affiliations:** 1Department of Oral Science, Graduate School of Medicine, Chiba University, 1-8-1 Inohana, Chuo-ku, Chiba-shi 260-8670, Japan; kawasakik@chiba-u.jp (K.K.); axga0845@chiba-u.jp (T.A.); nobuchit@chiba-u.jp (T.N.); nozaki.r@chiba-u.jp (R.N.); 2Department of Dentistry and Oral-Maxillofacial Surgery, Chiba University Hospital, 1-8-1 Inohana, Chuo-ku, Chiba-shi 260-8677, Japan; tomoakisaito@chiba-u.jp (T.S.); iyodam@chiba-u.jp (M.I.)

**Keywords:** platelet-activating factor receptor, cisplatin, ginkgolide B, oral squamous cell carcinoma, combination chemotherapy

## Abstract

**Simple Summary:**

The platelet-activating factor receptor (PAFR) is a key molecule that participates in intracellular signaling pathways. It is involved in cancer progression, but the detailed mechanism of its chemosensitivity is unknown. The purpose of the current study was to elucidate the mechanism regulating cisplatin (CDDP) sensitivity through PAFR functions in oral squamous cell carcinoma (OSCC). These results suggest that PAFR is a therapeutic target for modulating CDDP sensitivity in OSCC cells. In addition, we found that ginkgolide B (GB), a specific inhibitor of PAFR, enhanced both CDDP chemosusceptibility and apoptosis. Thus, GB may be a novel drug that could enhance combination chemotherapy with CDDP for OSCC patients.

**Abstract:**

The platelet-activating factor receptor (PAFR) is a key molecule that participates in intracellular signaling pathways, including regulating the activation of kinases. It is involved in cancer progression, but the detailed mechanism of its chemosensitivity is unknown. The purpose of the current study was to elucidate the mechanism regulating cisplatin (CDDP) sensitivity through PAFR functions in oral squamous cell carcinoma (OSCC). We first analyzed the correlation between PAFR expression and CDDP sensitivity in seven OSCC-derived cell lines based upon cell viability assays. Among them, we isolated 2 CDDP-resistant cell lines (Ca9-22 and Ho-1-N-1). In addition to conducting PAFR-knockdown (si*PAFR*) experiments, we found that ginkgolide B (GB), a specific inhibitor of PAFR, enhanced both CDDP chemosusceptibility and apoptosis. We next evaluated the downstream signaling pathway of PAFR in si*PAFR*-treated cells and GB-treated cells after CDDP treatment. In both cases, we observed decreased phosphorylation of ERK and Akt and increased expression of cleaved caspase-3. These results suggest that PAFR is a therapeutic target for modulating CDDP sensitivity in OSCC cells. Thus, GB may be a novel drug that could enhance combination chemotherapy with CDDP for OSCC patients.

## 1. Introduction

Oral squamous cell carcinoma (OSCC) is one of the most common cancers in the head and neck regions, constituting approximately 3% of all cancers [1]. Although progress has been made in recent years, the overall 5-year survival rate of OSCC patients remains unsatisfactory, standing at less than 50% [2].

In some cases, chemotherapy is an efficient adjuvant treatment for OSCC patients. However, the emergence of resistance to anti-cancer drugs hampers the curative effect to a large extent [3]. Cisplatin (CDDP) is a platinum-based anti-cancer drug used for a broad range of cancers. However, the severe side effects and frequent chemoresistance often limit its clinical application [4]. Therefore, understanding the molecular mechanisms of CDDP chemoresistance acquisition is critical and essential for improving the therapeutic outcome of OSCC patients.

Platelet activating factor (PAF), synthesized by various types of cells, is implicated in inflammation, carcinogenesis, and tumor metastasis [5,6]. PAF binds and induces biological activities through a unique 7-transmembrane G-protein-coupled receptor, the PAF-receptor (PAFR), which possesses exceptionally high affinity for its ligand [7,8,9]. The PAFR is expressed on the surface of various mammalian cells, including leukocytes, tissue macrophages and cancer cells [10,11]. PAFR expression directly regulates tumor growth via induction of systemic immunosuppressive effects and by positive feed-forward mechanisms. Upregulation of PAFR is also detected in primary tumors as well as tumors metastasizing to lymph nodes [12,13]. PAFR expression is positively correlated with advancing tumor stage, invasiveness, and poor prognosis in several types of cancer [12,13]. PAFR function is also closely related to cancer chemotherapy in epithelial carcinoma [14]. Therefore, we hypothesized here that PAFR regulates the effect of CDDP chemotherapy in OSCC patients.

In the current study, we found that PAFR expression status was involved in CDDP sensitivity in OSCC cells based upon *PAFR* knockdown experiments. In addition, we found that ginkgolide B (GB), an inhibitor of PAFR, increased the CDDP chemosusceptibility, suggesting that GB may be a novel drug that could enhance CDDP chemotherapy.

## 2. Materials and Methods

### 2.1. Ethics Statement

This study protocol has been approved by the Ethics Committee of the Graduate School of Medicine, Chiba University (approval number, 680).

### 2.2. Cells Lines

A total of 7 human OSCC-derived cell lines (HSC-2, HSC-3, HSC-3-M3, Ca9-22, Sa3, Ho-1-u-1, and Ho-1-N-1) were purchased from RIKEN BioResource Center (Tsukuba, Japan) and the Japanese Collection of Research Bioresources Cell Bank (Ibaraki, Japan). OSCC-derived cells were cultured as previously described [15]. We obtained human normal oral keratinocytes (HNOKs) from healthy young volunteer patients. HNOKs were cultured as previously described and used as normal controls [16].

### 2.3. mRNA Expression Analysis

PCR reaction conditions were previously described [15]. Primer 3Plus (online free software, http://primer3plus.com/, accessed on 16 April 2019) was used to design the primers. The sequences of the designed primers were as follows.: *PAFR* (forward, 5′-GACAGCATAGAGGCTGAGGC-3′; reverse 5′-TAGCCATTAGCAATGACCCC-3′) and glyceraldehyde-3-phospate dehydrogenase (*GAPDH*, forward, 5′-CATCTCTGCCCCCTCTGCTGA-3′; reverse, 5′-GGATGACCTTGCCCACAGCCT-3′). The normalization of the transcript levels of target genes was previously described [16].

### 2.4. Immunoblot Analysis

Proteins were extracted from cells using RIPA buffer (Nacalai Tesque, Kyoto, Japan) containing cOmplete ™ Protease Inhibitor Cocktail (Sigma-Aldrich, St. Louis, MO, USA) and the protein concentration was adjusted to 1 mg/mL. Immunoblotting was performed as previously described [16]. The primary antibodies used in the experiments were as follows: rabbit anti-PAFR, # bs-14730R-A750 (Bioss. Inc., Woburn, MA, USA), 1:200; rabbit anti-Erk1/2 # 9102 (Cell Signaling Technology, Beverly, MA, USA), 1:1000; rabbit anti-p-Erk1/2 # 4370 (Cell Signaling Technology), 1:1000; rabbit anti-Akt # 4691 (Cell Signaling Technology), 1:1000; rabbit anti-p-Akt #4060 (Cell Signaling Technology), 1:1000; rabbit anti-Caspase-3 #9662 (Cell Signaling Technology), 1:1000; rabbit anti-Cleaved caspase-3 #9661 (Cell Signaling Technology), 1:1000; and mouse anti-α tubulin # sc-5286 (Santa Cruz Biotechnology, Shanghai, China).

### 2.5. Cellular Proliferation Assay

The cells were treated with the indicated concentrations of CDDP (FUJIFILM Wako Pure Chemical Corporation, Osaka, Japan) and/or GB, a PAFR inhibitor (Selleck Chemicals, Houston, TX, USA) for the indicated time periods. Cell viability was determined as previously described [17]. Half-maximal inhibitory concentrations (IC_50_) values were calculated from semi-logarithmic dose-response curves by linear interpolation.

### 2.6. Transfection of PAFR siRNA

Stealth *PAFR* siRNAs (si*PAFR*) (HSS108752, HSS108753, HSS183786) (Thermo Fisher Scientific, Waltham, MA, USA) and control siRNAs (control) (Thermo Fisher Scientific) were used to transfect OSCC cells. The introduction of siRNA was conducted as previously described [18].

### 2.7. Apoptosis Assays

The FITC Annexin V Apoptosis Detection Kit I (Becton-Dickinson, Franklin Lakes, NJ, USA) was used to measure apoptosis. Briefly, cells were treated with trypsin and collected. The collected cells were washed, and the cell suspension was adjusted to 1 × 10^5^/100 µL. Annexin V-FITC and PI were added to 100 µL of cell suspension and assayed according to the manufacturer’s instructions.

### 2.8. Statistical Analysis

The data were analyzed using an unpaired *t*-test or one-way analysis of variance (ANOVA), followed by a post hoc Tukey’s test (ANOVA with Tukey’s multiple comparison test). All statistical analyses were performed with Microsoft Excel (Microsoft, Redmond, WA, USA). The data are expressed as the mean ± standard error of the mean.

## 3. Results

### 3.1. PAFR Expression in OSCC Cells

To analyze the expression status of *PAFR*, the seven OSCC cell lines and HNOKs were subjected to qRT-PCR and immunoblot analyses. *PAFR* mRNA expression was up-regulated significantly (ANOVA with Tukey’s multiple comparison test, *p* < 0.05.) in the four OSCC cells (except HSC-3, Sa3, and Ho-1-u-1) compared with the HNOKs (Figure 1A). PAFR protein expression was also significantly upregulated in six OSCC cell lines compared with the HNOKs (Figure 1B).

### 3.2. CDDP Sensitivity of OSCC Cells

To investigate the susceptibility of OSCC cells (Ca9-22, Ho-1-N-1, HSC-2, HSC-3-M3, Sa3, HSC-3, and Ho-1-u-1) to CDDP, we assessed cell viability after treatment with the drug (Figure 2A). Figure 2B shows the IC_50_ of OSCC cells for CDDP. These results indicated that Ca9-22 was the most resistant and Ho-1-u-1 was the most sensitive to CDDP. From the results of Figure 1 and Figure 2, the resistance to CDDP in OSCC cells had positive correlations to PAFR expression status. Considering these results, Ho-1-N1 and Ca9-22 cells were selected for further experiments.

### 3.3. Effect of PAFR Knockdown on Cell Proliferation

The expression levels of *PAFR* mRNA and PAFR protein in si*PAFR*-transfected cells decreased significantly (Unpaired *t*-test, *p* < 0.001; n.s., no significant difference) compared with the control cells (Figure 3A–D). Based upon cellular proliferation assays, there was no significant effect of si*PAFR* transfection on cell proliferation (Figure 3E,F).

### 3.4. Effect of PAFR Knockdown on CDDP Sensitivity

We investigated CDDP susceptibility in si*PAFR*-transfected cells (Figure 4A,B). After si*PAFR* transfection, cells were treated with CDDP (0–1000 μM) for 48 h. si*PAFR*-transfected cells had higher susceptibility to CDDP than control cells (Figure 4C,D; Unpaired *t*-test; *, *p* < 0.01; **, *p* < 0.001). These data suggested that PAFR expression was involved in the regulation of CDDP susceptibility in OSCC cells. To further investigate apoptosis of OSCC cells after CDDP treatment, we performed flow cytometric analysis. The apoptosis frequency of si*PAFR*-transfected OSCC cells was higher than that of the control group (Figure 5A,B; Unpaired *t*-test; *, *p* < 0.05; **, *p* < 0.01). Since the activation of ERK and Akt signaling pathways are common attributes of apoptosis or survival [19], we investigated ERK and Akt activation in si*PAFR*-transfected OSCC cells after treatment with CDDP. We also investigated cleaved caspase-3 to confirm apoptosis. Decreased ERK and Akt phosphorylation was observed in si*PAFR*-transfected Ca9-22 and Ho-1-N-1 cells after treatment with CDDP. In addition, highly cleaved caspase-3, an apoptosis marker, was observed in both Ca9-22 and Ho-1-N-1 cells after treatment with CDDP (Figure 5C).

### 3.5. Effect of GB on Cell Proliferation

We examined the effect of GB on cell growth in OSCC cells by cellular proliferation assay. After treatment with 200 µM GB for the indicated duration, no effect on cell proliferation of OSCC cells was found (Figure 6A,B).

### 3.6. Effect of GB on CDDP Sensitivity

First, we assessed the capability of GB as a specific inhibitor of PAFR and found that PAF-induced IL-1β expression significantly decreased after treatment with GB (Unpaired *t*-test, *p* < 0.001, Appendix A). Cellular viability assay, flow cytometry and immunoblot analyses were performed to investigate the effect of CDDP plus GB combination therapy on OSCC cells (Figure 7A,B). The CDDP sensitivity of OSCC cells increased in a GB dose-dependent manner (Figure 7C,D, Unpaired *t*-test, *p* < 0.05). The apoptosis frequency of the cells treated with CDDP plus GB treatment was higher than that of treatment with CDDP alone (Figure 8A,B; Unpaired *t*-test; *, *p* < 0.05; **, *p* < 0.01; n.s., no significant difference). We next investigated downstream signaling of PAFR after CDDP and GB treatments. In both Ca9-22 and Ho-1-N1 cells, reduced phosphorylation of ERK and Akt and highly cleaved caspase-3 were observed in cells treated with CDDP plus GB combination therapy (Figure 8C).

## 4. Discussion

CDDP is a widely used chemotherapeutic drug used for cancer, including OSCC [20] however, the acquisition of CDDP resistance markedly restricts its application. Since the severe side effects and chemoresistance are important factors for CDDP chemotherapy for OSCC patients [21], an agent that reduces such complications is required. In the present study, we found that the suppression of *PAFR* by siRNA (Figure 4 and Figure 5) and GB (Figure 7 and Figure 8) enhanced the CDDP sensitivity of OSCC cells. Furthermore, our data indicated that ERK and Akt signaling, downstream of PAFR, may be key pathways in CDDP treatment (Figure 5 and Figure 8).

In various cancers, PAFR overexpression accelerates cell proliferation, migration, and invasion relative to control cells [12,13,22,23,24,25,26,27,28]. In addition, high-PAFR tumors showed significantly decreased overall survival compared to low-PAFR tumors [20,21]. These studies suggested that tumors’ PAFR expression levels are closely related to not only tumor progression but also cancer prognosis. The PAFR signaling pathway has been shown to activate ERK and Akt, both of which mediate important signals for cell proliferation, survival, and differentiation in several types of cancer cells [29,30]. Similarly to the previous data [31,32], our study demonstrated that the inducible activation of the ERK and Akt pathways is associated with chemotherapy resistance in OSCC cells (Figure 5C and Figure 8C).

Ginkgolides have been isolated from *Ginkgo biloba*, a Chinese herb, and used in traditional Chinese medicine for thousands of years [33]. Currently, ginkgolides are used for analgesia, suppression of wheezing, and treatment of cerebrovascular disease, coronary artery disease, and hypertension [34]. Ginkgolides, including ginkgolide A, B, C, J, K, L, and M were found to be specific and selective antagonists of PAFR. Of them, GB has the most potent inhibitory effect on PAFR [35,36]. GB has many beneficial characteristics, such as anti-inflammatory properties, as well as anti-allergic, antioxidant, and neuroprotective effects. Thus, it offers significant therapeutic actions in many diseases [37]. To date, no serious side effects directly attributable to GB have been reported [38,39].

We focused on GB as a selective inhibitor of PAFR, and we investigated the effect of GB on CDDP sensitivity in OSCC. To clarify whether GB enhances CDDP sensitivity to Ca9-22 and Ho-1-N-1 cell lines, cell viability assays, flow cytometric analyses, and immunoblot analyses were performed. The results showed that CDDP combined with GB reduced cell viability and increased cell apoptosis, while GB alone had no such impact (Figure 7 and Figure 8). Side effects of anticancer drugs, especially CDDP, have become a serious problem. Our data indicated that GB not only enhanced the sensitivity to CDDP but also achieved the same efficacy with lower doses of CDDP for the patients of several types of cancer.

## 5. Conclusions

These results suggest that PAFR is involved in the regulation of CDDP sensitivity in OSCC. In addition, GB increased CDDP sensitivity through its effects on the PAFR signaling pathways. Our results suggest that GB may have therapeutic efficacy when used in combination with CDDP in OSCC.

## Figures and Tables

**Figure 1 cancers-13-06299-f001:**
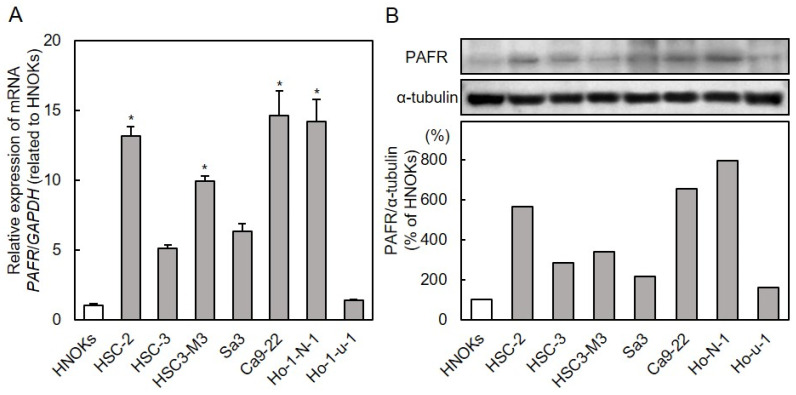
PAFR expression in OSCC cells. (**A**) Quantification of *PAFR* mRNA expression in OSCC cells by qRT-PCR analysis. *PAFR* mRNA expression was significantly upregulated in four OSCC cell lines compared to HNOKs (ANOVA with Tukey’s multiple comparison test; *, *p* < 0.05). (**B**) Representative immunoblot analyses of PAFR protein expression. PAFR protein expression was upregulated in six OSCC cell lines compared to HNOKs. Densitometry data were normalized to α-tubulin protein levels. Values are expressed as percentages of HNOKs. Detailed information about the Western blotting can be found at Appendix A and Appendix A. OSCC, oral squamous cell carcinoma; HNOKs, human normal oral keratinocytes.

**Figure 2 cancers-13-06299-f002:**
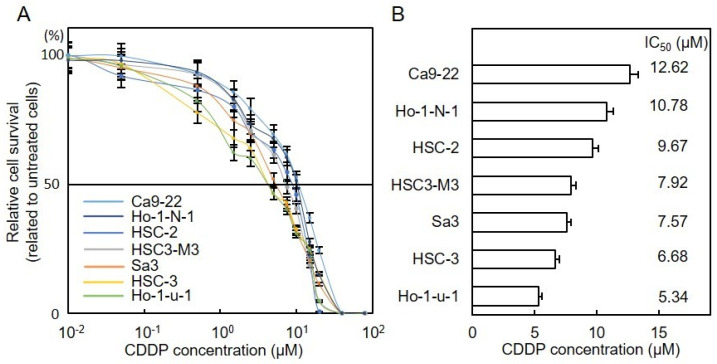
CDDP sensitivity of OSCC cells. (**A**) OSCC cells were seeded in 96-well plates at a density of 1 × 10^4^ viable cells/well and treated with CDDP (0.01–1000 µM) for 48 h, followed by assessment of cellular viability. Values are expressed as the mean ± standard error of the mean in relation to untreated cells. (**B**) Half-maximal inhibitory concentrations (IC_50_) of CDDP in OSCC cells. The most resistant IC_50_ to CDDP, 12.62 µM (Ca9-22); the most sensitive IC_50_ to CDDP, 5.34 µM (Ho-1-u-1). CDDP, cisplatin; OSCC, oral squamous cell carcinoma.

**Figure 3 cancers-13-06299-f003:**
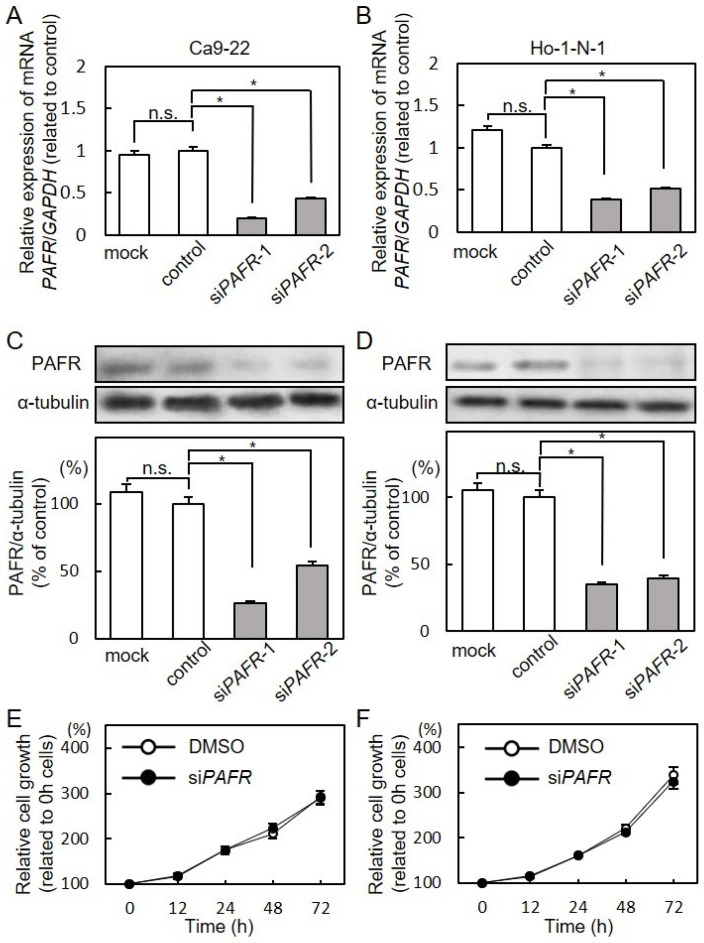
Effect of *PAFR* knockdown on cell proliferation. (**A**,**B**) Expression of *PAFR* mRNA was significantly reduced by si*PAFR* transfection compared with control (Ca9-22 (**A**) and Ho-1-N-1 (**B**); Unpaired *t*-test; *, *p* < 0.001; n.s., no significant difference). (**C**,**D**) Expression of PAFR protein was markedly reduced by si*PAFR* transfection compared with control (Ca9-22 (**C**) and Ho-1-N-1 (**D**); Unpaired *t*-test; *, *p* < 0.001; n.s., no significant difference). Densitometry data were normalized to α-tubulin protein levels. Values are expressed as percentages of control. (**E**,**F**) To determine the effect of si*PAFR* on cellular proliferation, si*PAFR*-transfected cells were seeded in 96-well plates at a density of 1 × 10^4^ cells/well, followed by an assessment of cellular viability. The cells were counted at the indicated times. Cell proliferation after si*PAFR* transfection was not significantly different from controls (Ca9-22 (**E**) and Ho-1-N-1 (**F**)). Detailed information about the Western blotting can be found at Appendix A and Appendix A.

**Figure 4 cancers-13-06299-f004:**
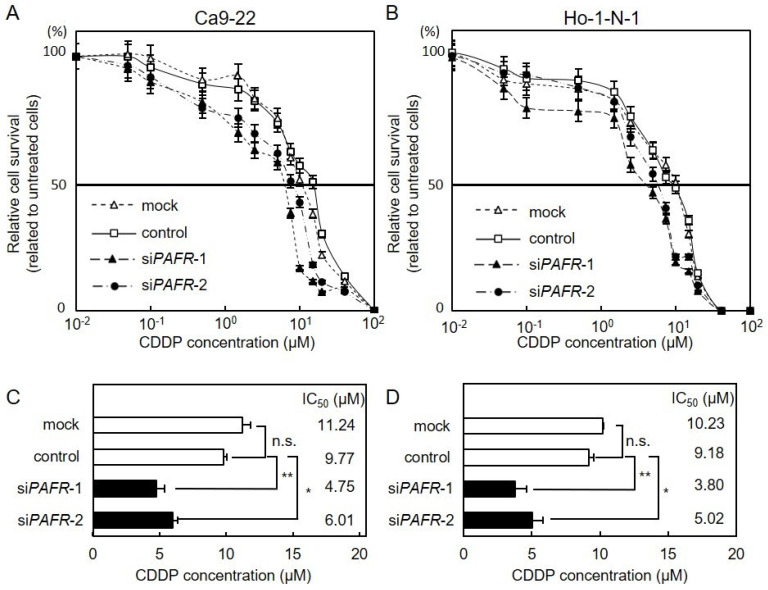
Effect of *PAFR* knockdown on CDDP sensitivity. (**A**,**B**) si*PAFR*-transfected cells were seeded in 96-well plates at a density of 1 × 10^4^ cells/well and treated with CDDP (0.01–1000 µM) for 48 h, followed by a cellular viability assay. Values are expressed as the mean ± standard error of the mean in relation to untreated cells (Ca9-22 (**A**) and Ho-1-N-1 (**B**)). (**C**,**D**) IC_50_ of CDDP in si*PAFR*-transfected cells. A significant decrease in IC_50_ was observed in si*PAFR*-transfected cells compared with control (Ca9-22 (**C**) and Ho-1-N-1 (**D**); Unpaired *t*-test; *, *p* < 0.01; **, *p* < 0.001; n.s., no significant difference). CDDP, cisplatin.

**Figure 5 cancers-13-06299-f005:**
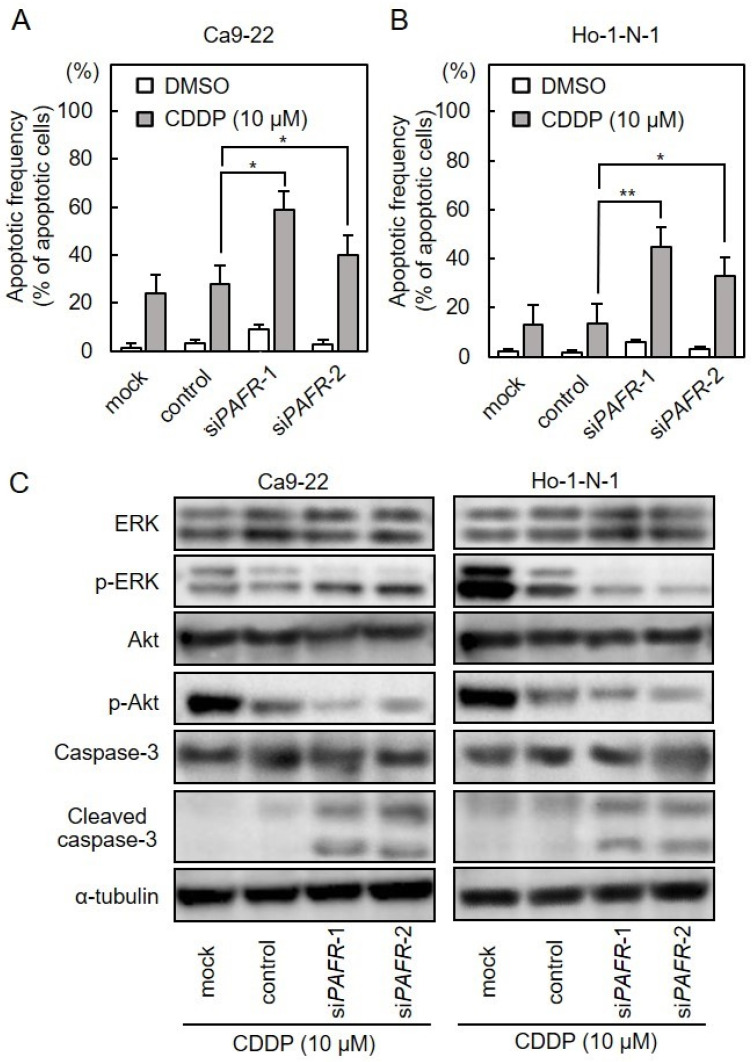
Effects of *PAFR* knockdown on apoptosis and its downstream pathways. (**A**,**B**) Flow cytometric assays were used to assess apoptosis frequency in si*PAFR*-transfected cells after CDDP treatment. Each value represents mean values obtained from three separate experiments. Asterisks indicate statistical significance (Ca9-22 (**A**) and Ho-1-N-1 (**B**); Unpaired *t*-test; *, *p* < 0.05; **, *p* < 0.01). (**C**) Immunoblot analysis of the phosphorylation of ERK, phosphorylation of Akt and cleaved caspase-3. Decreased ERK and Akt phosphorylation was observed in si*PAFR*-transfected cells in both Ho-1-N-1 and Ca9-22 after treatment with CDDP. In addition, highly cleaved caspase-3, an apoptosis marker, was observed in both Ca9-22 and Ho-1-N-1 after treatment with CDDP. α-tubulin protein levels were used as loading control. Detailed information about the Western blotting can be found at Appendix A and Appendix A. CDDP, cisplatin.

**Figure 6 cancers-13-06299-f006:**
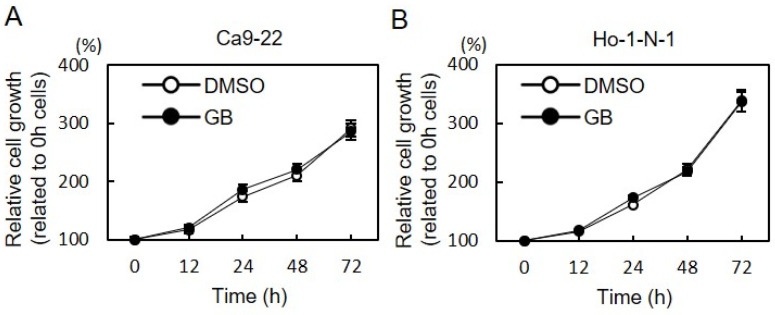
Effect of GB on cell proliferation. (**A**,**B**) To determine the effect of GB on cellular proliferation, OSCC cells were seeded in 96-well plates at a density of 1 × 10^4^ cells/well and treated with GB (200 µM) or DMSO for 24 h, followed by cellular viability assay. The cells were counted at the indicated times. There was no significant difference in OSCC cell proliferation after treatment with GB (Ca9-22 (**A**) and Ho-1-N-1 (**B**)). GB, ginkgolide B; OSCC, oral squamous cell carcinoma.

**Figure 7 cancers-13-06299-f007:**
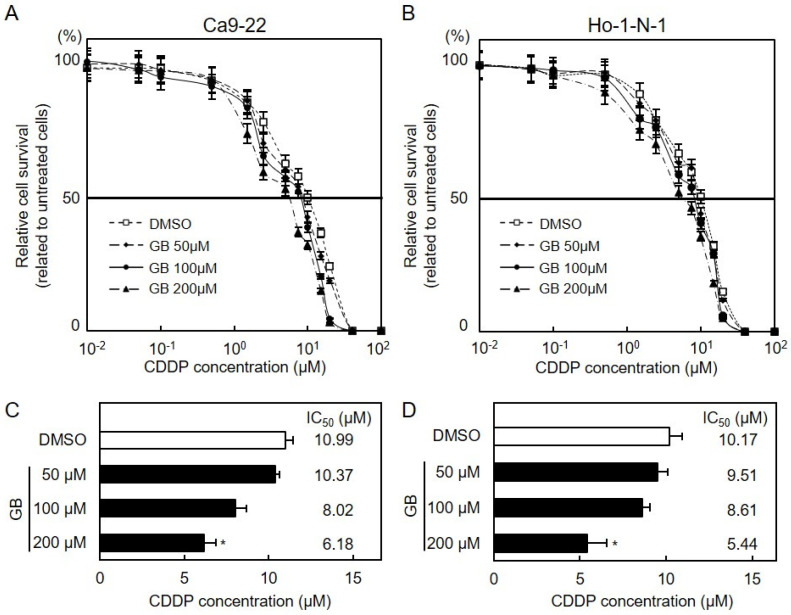
Effect of GB on CDDP sensitivity. (**A**,**B**) OSCC cells were seeded in 96-well plates at a density of 1 × 10^4^ cells/well and treated with GB (0, 50, 100, and 200 µM) for 24 h. Next, OSCC cells were treated with CDDP (0.01–1000 µM) for 48 h, followed by a cellular viability assay. Values are expressed as the mean ± standard error of the mean in relation to untreated cells (Ca9-22 (**A**) and Ho-1-N-1 (B)). (**C**,**D**) IC_50_ of CDDP in GB-treated cells (Ca9-22 (**C**) and Ho-1-N-1 (**D**)). Asterisks indicate statistical significance compared with DMSO (Ca9-22 (**C**) and Ho-1-N-1 (**D**); Unpaired *t*-test; *, *p* < 0.05). CDDP, cisplatin; GB, ginkgolide B; OSCC, oral squamous cell carcinoma.

**Figure 8 cancers-13-06299-f008:**
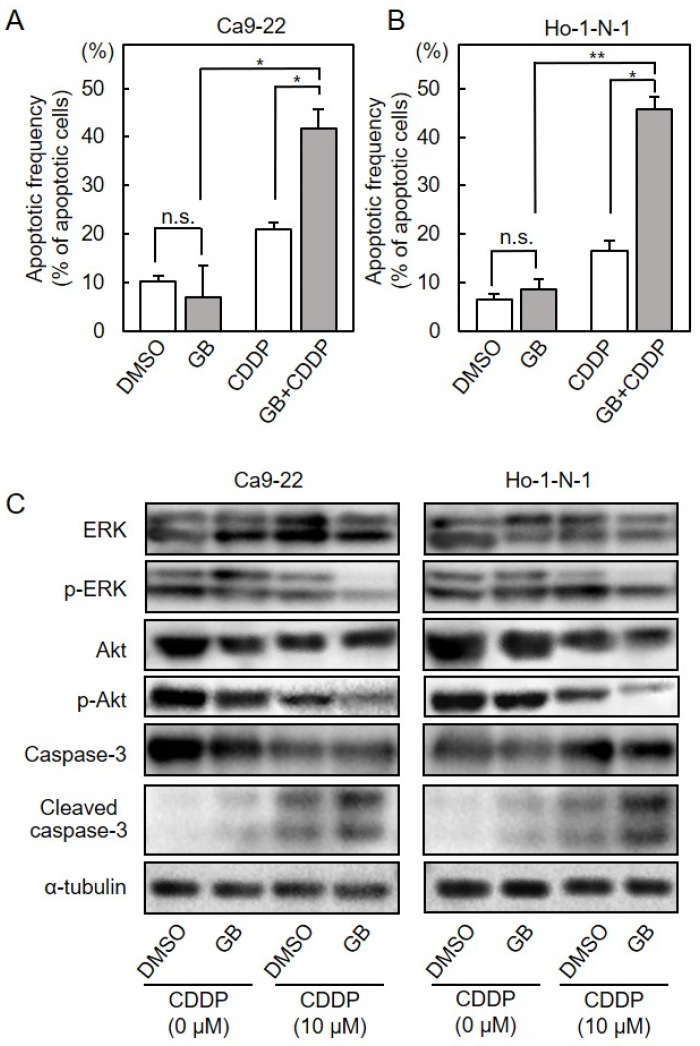
Effects of GB on CDDP sensitivity. (**A**,**B**) Flow cytometric assays of apoptosis in OSCC cells after treatment with GB, CDDP, and CDDP plus GB. Each represents mean values obtained from three separate experiments. Asterisks indicate statistical significance (Ca9-22 (**A**) and Ho-1-N-1 (**B**); Unpaired *t*-test; *, *p* < 0.05; **, *p* < 0.01; n.s., no significant difference). (**C**) Immunoblot analysis of the phosphorylation of ERK, phosphorylation of Akt, and cleaved caspase-3. In both Ca9-22 and Ho-1-N1, reduced phosphorylation of ERK and Akt and highly cleaved caspase-3 were observed in cells treated with CDDP plus GB combination therapy. α-tubulin protein levels were used as loading control. Detailed information about the Western blotting can be found at Appendix A and Appendix A. CDDP, cisplatin; GB, ginkgolide B; OSCC, oral squamous cell carcinoma.

## Data Availability

All relevant data are within the paper and its Appendix A.

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
