# Peer review of "Ginkgolide B Regulates CDDP Chemoresistance in Oral Cancer via the Platelet-Activating Factor Receptor Pathway"

_cancers, 2021, doi:10.3390/cancers13246299_

Round 1

Reviewer 1 Report

Thank you for your responses. I have no further questions. 

Author Response

Responses to the Reviewer 1:

We thank the Reviewer 1 for his/her careful and comprehensive evaluation of our manuscript. We appreciate the comments and suggestions for improvement.

Review comments:

Thank you for your responses. I have no further questions.

Response:

Thank you very much for your special consideration.

Reviewer 2 Report

The authors provided interesting data stating that GB as a PAFR inhibitor increases CDDP sensitivity in oral squamous cells that combination treatment of GB and CDDP significantly induced cell apoptosis and reduced PAFR activation. 

The western data of p-Akt and p-Erk expression (figure 8C) of GB treatment alone did not show inhibitory effects, but PAFR expression data (figure 3) and IL-1b expression data (figure S5) provided evidence that GB inhibited PAFR expression and function. Please add few words in your text to state that figure S5 data supported the inhibitory effects of GB on PAFR function.

Author Response

Responses to the Reviewer 2:

We thank the Reviewer 2 for his/her careful and comprehensive evaluation of our manuscript. We appreciate the comments and suggestions for improvement. We revised the manuscript as indicated below to address the points raised by the Reviewer 2.

Review comments:
Q1.

The authors provided interesting data stating that GB as a PAFR inhibitor increases CDDP sensitivity in oral squamous cells that combination treatment of GB and CDDP significantly induced cell apoptosis and reduced PAFR activation.

The western data of p-Akt and p-Erk expression (figure 8C) of GB treatment alone did not show inhibitory effects, but PAFR expression data (figure 3) and IL-1b expression data (figure S5) provided evidence that GB inhibited PAFR expression and function. Please add few words in your text to state that figure S5 data supported the inhibitory effects of GB on PAFR function.

Responses (Q1):

We thank the Reviewer for this comment. According to the reviewer’s suggestion, we have added the comments in the Results section as follows:

Additional comment

‘First, we assessed the capability of GB as a specific inhibitor of PAFR, and found that PAF-induced IL-1β expression was significantly decreased after treatment with GB (Unpaired t-test, P < 0.001, Figure S5).’ (page, 8; lines, 198-200)

This manuscript is a resubmission of an earlier submission. The following is a list of the peer review reports and author responses from that submission.

Round 1

Reviewer 1 Report

In the submitted study, “Ginkgolide B regulates CDDP chemoresistance in oral cancer via the platelet-activating factor receptor pathway”. is nicely designed and firstly analysed the mechanism regulating cisplatin sensitivity through PAFP functions in oral cancer  and then it foward to verify that GB could enhance chemosusceptibility to cisplatin, constituting a promising approach with putative clinical relevance.

The topic is interesting and I believe the manuscript reflects a well design study. However, authors should clarify some aspects of the work.

I)             Introduction: A clear hypothesis should be provided, based on the identified knowledge gap and aims of the work should be clearly presented.

II)            In figures 1 and 2 for example, in Y axis is depicted relative expression of mRNA or relative cell survival. Authors need to clarify "relative" to which condition they refer to? This is something that occurs through the all figures presented.

III)        Authors need to detail the statistical test performed in each figure and clarify the * P values significant difference value and explain the difference to each condition

IV)           Statistics: Authors should clarify which test they have used to evaluate the normality of data.  Also, authors refer the use of Student’s t test: It is paired or unpaired test? I also recommend the use of multicomparison test to evaluate differences in results obtained between the several tumour cell lines used in figure 1.

V) Discussion: Integration of data obtained in the study with current knowledge in oral cancer as well as in other cancer types is missing and is recommended. From a reader point of view this will help to understand the importance of the work presented and the scientific achievements.

Reviewer 2 Report

PAFR activation induced systematic immunosuppression directly regulate tumor progression. The authors stated that inhibition of PAFR in oral squamous cell carcinoma derived cell lines enhances CDDP chemosensitivity. The authors aim to use GB as PAFR inhibitor in this study but the immunoblot data of ginkgolide B treatment is not very convincing according to the figure provided (Figure 8C).

In general, the authors provided some interesting and useful information on this topic, but its necessary to confirm the GB inhibitory effects on PAFR and make necessary revisions.    
